# Excess Heme Promotes the Migration and Infiltration of Macrophages in Endometrial Hyperplasia Complicated with Abnormal Uterine Bleeding

**DOI:** 10.3390/biom12060849

**Published:** 2022-06-19

**Authors:** Lu-Yu Ruan, Zhen-Zhen Lai, Jia-Wei Shi, Hui-Li Yang, Jiang-Feng Ye, Feng Xie, Xue-Min Qiu, Xiao-Yong Zhu, Ming-Qing Li

**Affiliations:** 1NHC Key Lab of Reproduction Regulation, Shanghai Institute of Planned Parenthood Research, Hospital of Obstetrics and Gynecology, Fudan University, Shanghai 200080, China; 19211250036@fudan.edu.cn; 2Laboratory for Reproductive Immunology, Institute of Obstetrics and Gynecology, Hospital of Obstetrics and Gynecology, Fudan University, Shanghai 200080, China; 21111250012@fudan.edu.cn (Z.-Z.L.); 21111250018@m.fudan.edu.cn (J.-W.S.); yanghuili7976@fckyy.org.cn (H.-L.Y.); 3Institute for Molecular and Cell Biology, Agency for Science, Technology and Research, Singapore 138632, Singapore; jiangfeng_ye@hotmail.com; 4Medical Center of Diagnosis and Treatment for Cervical and Intrauterine Diseases, Obstetrics and Gynecology Hospital of Fudan University, Shanghai 200011, China; 5Clinical Research Center, Hospital of Obstetrics and Gynecology, Fudan University, Shanghai 200011, China; 6Department of Gynecology, Hospital of Obstetrics and Gynecology, Fudan University, Shanghai 200011, China; 7Shanghai Key Laboratory of Female Reproductive Endocrine Related Diseases, Hospital of Obstetrics and Gynecology, Fudan University, Shanghai 200011, China

**Keywords:** endometrial hyperplasia, immune cells, macrophages, heme, abnormal uterine bleeding, HO-1

## Abstract

In patients, endometrial hyperplasia (EH) is often accompanied by abnormal uterine bleeding (AUB), which is prone to release large amounts of heme. However, the role of excess heme in the migration and infiltration of immune cells in EH complicated by AUB remains unknown. In this study, 45 patients with AUB were divided into three groups: a proliferative phase group (n = 15), a secretory phase group (n = 15) and EH (n = 15). We observed that immune cell subpopulations were significantly different among the three groups, as demonstrated by flow cytometry analysis. Of note, there was a higher infiltration of total immune cells and macrophages in the endometrium of patients with EH. Heme up-regulated the expression of heme oxygenase-1 (HO-1) and nuclear factor erythroid-2-related factor 2 (Nrf2) in endometrial epithelial cells (EECs) in vitro, as well as chemokine (e.g., CCL2, CCL3, CCL5, CXCL8) levels. Additionally, stimulation with heme led to the increased recruitment of THP-1 cells in an indirect EEC-THP-1 co-culture unit. These data suggest that sustained and excessive heme in patients with AUB may recruit macrophages by increasing the levels of several chemokines, contributing to the accumulation and infiltration of macrophages in the endometrium of EH patients, and the key molecules of heme metabolism, HO-1 and Nrf2, are also involved in this regulatory process.

## 1. Introduction

Endometrial hyperplasia is [1] one of the most common gynecological diseases in the world, impacting many women’s lives [2]. EH is a morphological endometrial change in which the ratio of endometrial stroma and glands is significantly higher than that in the normal endometrium [3]. The high-risk factors for EH are highly similar to those associated with endometrial cancer (EC) [4], which is also considered to be one of the precancerous lesions that most commonly develop into EC. Abnormal endometrial hyperplasia is generally not easy to detect when asymptomatic, and patients generally have abnormal menstruation—that is, abnormal uterine bleeding. Therefore, research on the pathogenesis of EH is of great significance for the early prevention of EC.

The occurrence of EH is related to a variety of factors. Immune factors are also involved. It has been reported that an imbalance between estrogen and progesterone may lead to the abnormal proliferation of the endometrial glandular epithelium. In addition to hormones, leukocytes producing molecules including cytokines (interleukin-1β (IL-1β), IL-6, and tumor necrosis factor (TNF)-α), adhesion molecules and growth factor network regulation have also been found to play an important role in the disease progression of EH [4,5]. In addition, immune cells such as natural killer cells, T cells, macrophages and neutrophils have been confirmed to be closely related to the occurrence of EH. In particular, macrophages are considered to play key roles in the occurrence and development of many gynecological diseases, due to their anti-inflammatory and pro-inflammatory properties. However, the overall subpopulation characteristics of these immune cells in the endometrium and their infiltrating and resident mechanisms remain unclear.

Abnormal uterine bleeding (AUB) is a common condition [6]. It may first occur at adolescence, when menarche happens [7], but is notably prevalent in women of reproductive age [8]. Approximately 3–30% of reproductive-age women suffer from this condition [9]. AUB is described as any difference from normal menstrual bleeding [10]. However, the use of terminology and documentation of etiology of AUB show a lack of agreement. In 2007 and 2011, the International Federation of Gynecology and Obstetrics [9] created Systems 1 and 2 for AUB, respectively, providing clear terminology and nomenclature [9]. FIGO System 1 identified four defined criteria for menses: frequency, duration, regularity and volume. FIGO System 2 systematically defined the potential causes of AUB, summarized as the acronym “PALM-COEIN” (P—polyp[s], A—adenomyosis, L—leiomyoma, M—malignancy, C—coagulopathy, O—ovulatory dysfunction, E—endometrial disorders, I—iatrogenic and N—not yet classified) [11]. AUB is often the main manifestation in patients with endometrial dysplasia. However, the changes in immune cell subsets in AUB caused by endometrial dysplasia and other causes are not clear.

There might be a large amount of heme release in the uterus of a patient with abnormal uterine bleeding. Heme is produced from the breakdown of hemoglobin in the blood and is thought to be responsible for many types of inflammatory and oxidative damage. Under normal physiological conditions, heme is catabolized by the key enzyme heme oxygenase-1 (HO-1). Under pathological conditions, the body’s ability to process heme reaches saturation, resulting in the excessive accumulation of heme, leading to body damage and disease [12]. Existing studies have shown that heme may affect the function of various immune cells, including macrophages, and promote the recruitment and activation of immune cells such as neutrophils and macrophages by regulating some chemokines and adhesion molecules, and it may ultimately affect the occurrence and development of diseases [12,13]. However, the possible roles of heme in EH, and whether it plays a role by regulating the infiltration and function of immune cells such as macrophages, are currently unclear.

In this study, we investigated the changes in immunocyte subsets in different phases of the menstrual cycle in women with AUB and in the endometrial tissue of patients with EH, as well as the role of heme in the migration and recruitment of macrophages by endometrial cells.

## 2. Materials and Methods

### 2.1. Patients

The study was approved by the Human Research Ethics Committee of the Obstetrics & Gynecology Hospital of Fudan University, and carried out at the Obstetrics & Gynecology Hospital of Fudan University. Informed consent was obtained from all participants. Forty-five cases who underwent dilatation and curettage due to abnormal uterine bleeding and reported to display the “proliferative phase of the menstrual cycle”, “secretory phase of the menstrual cycle” or “endometrial hyperplasia” according to pathology results were recruited for the study. Based on the pathology reports, three groups were created: proliferative phase (n = 15), secretory phase (n = 15) and endometrial hyperplasia (n = 15) (Table 1). All the samples were transported to the laboratory on ice in Dulbecco’s modified Eagle’s medium (DMEM)/F-12 (Gibco) for further study.

### 2.2. Sample Preparation and Immunocyte Isolation

To wash away the remaining blood, the endometrial tissue samples were washed with ice-cold phosphate-buffered saline (PBS). Then, the tissue samples were minced into 2 mm pieces on ice and subsequently digested with 10% collagenase type IV (0.1%; Sigma, SL, MO, USA) for 35 min at 37 °C, with constant agitation to isolate immunocytes. To remove cellular debris, the tissue pieces were later filtered through a 70 μm nylon cell strainer (Falcon, Corning, NY, USA), followed by centrifugation at 400× *g* for 8 min for the collection of all types of cells. Then, 15 mL red blood cell lysis buffer was added into the pellet for 15 min on ice for the removal of remaining erythrocytes. Later, the cells were resuspended in DMEM/F-12 containing 10% fetal bovine serum (FBS; Hyclone, Logan, UT, USA), plated on culture flasks and incubated in a humidified incubator with 5% CO_2_ at 37 °C. Finally, after 24 h, the supernatant was removed and centrifuged at 400× *g* at 4 °C for flow cytometry analysis.

### 2.3. Flow Cytometry Analysis (FCM)

FCM was used to identify different subsets of immunocytes. All antibodies were from Biolegend, San Diego, CA, USA (detailed antibody information listed in Table 2). Different antibodies aimed at different cell surface markers. All antibodies (5 μL separately) were used for staining at room temperature for 30 min in the dark. In the flow cytometry, Human Trustain FcX (422301, Biolegend) was first used to block Fc receptors. Later, cells were washed twice and resuspended in PBS for FCM analysis. Samples were analyzed with a CytoFLEX flow cytometer (Beckman Coulter, Inc., Brea, CA, USA) using FlowJo software (version 10.07, Becton Dickinson, Inc., Franklin Lakes, NJ, USA).

### 2.4. Cells

The human endometrium epithelial cell line (EEC, WHELAB C1225) was kindly provided by SHANGHAI WHELAB BIOSCIENCE LIMITED, and U937 (human monocyte cell line) cells were purchased from the American Type Culture Collection. EEC and U937 cells were cultured in MEM or RPMI 1640 (HyClone Laboratories, Logan, UT, USA) containing 10% FBS (Gibco Cell Culture, Carlsbad, CA, USA) and 1% antibiotic–antimycotic solution (Gibco Cell Culture, Carlsbad, CA, USA), respectively.

### 2.5. The Treatment of Heme and HO-1 Inhibitor and Nrf2 Inhibitor

EEC cells were treated with 0, 12.5 or 25 μM heme (H8130, Solarbio, Beijing, China), and the relevant cells were collected after 48 h. EEC cells were treated with HO-1 inhibitor (Zinc Protoporphyrin, 5 μM, HY-101193, MedChem Express, NJ, USA) or Nrf2 inhibitor (ML385, 5 μM, HY-100523, MedChem Express, NJ, USA), and the relevant cells were collected after 24 h.

### 2.6. Polymerase Chain Reaction (PCR)

Total RNA was extracted from cells using RNAiso Plus reagent (TaKaRa Biotechnology, Kusatsu, Japan), according to the manufacturer’s protocol. Total RNA was reverse-transcribed into first-stand cDNA (RR036A, TaKaRa Biotechnology), following the manufacturer’s protocol. cDNA was subsequently amplified with specific primers (detailed sequences listed in Table 3) (Sangon Biotech, Shanghai, China). The PCR reaction system (10 μL) was as follows: 1 μL cDNA + 5 μL PCR mix + 0.4 μL pre-primer + 0.4 μL post-primer + 3.2 μL RNase Free ddH_2_O. PCR was performed using the following conditions: 95 °C 30 s→(95 °C 5 s→60 °C 34 s) × 40 Cycle→95 °C for 15 s→60 °C for 1 min→95 °C for 15 s. Data were analyzed using the 2^−ΔΔCt^ method.

### 2.7. Integration Analysis of the Protein–Protein Interaction (PPI) Network

The STRING database (available online: http://string-db.org, 10 October 2021) was used for protein–protein interaction network prediction.

### 2.8. Chemotaxis Assay

We collected EEC cells and seeded them in 24-well plates. EEC cells were treated with 0, 12.5 and 25 μM heme for 48 h, and the relevant cells were collected and seeded in the lower chamber of the transwell. We collected U937 cells, stained them with Cell Tracker Red CMTPX dye, seeded them in the upper chamber of the transwell and co-cultured them with EEC cells. After 24 h of chemotaxis, the chemotaxis of macrophages was photographed using a fluorescence microscope.

### 2.9. Statistical Analysis

For the three groups, a one-way ANOVA with a Bonferroni multiple-comparisons test was used for continuous variables that fit a normal distribution, and the results were presented as mean ± SEM. A Kruskal–Wallis H test with Dunn’s multiple-comparisons test was used for continuous variables that fit a non-normal distribution, and the results were presented as the median and interquartile range. In addition, continuous variables were analyzed by t test for normal data or Mann–Whitney U test for non-normally distributed data between two groups. All analyses were conducted using the SPSS 20.0 Statistical Package for the Social Sciences software. A statistically significant difference was considered at *p* < 0.05.

## 3. Results

### 3.1. Baseline Characteristics

The characteristics of the women in this study are shown in Table 1. All the women had the clinical symptoms of AUB. Based on the pathological diagnosis of the endometrium, participants were divided into three groups: (1) proliferative phase of the menstrual cycle, (2) secretory phase of the menstrual cycle and (3) EH. The age and endometrial thickness of participants were similar in the three groups. Moreover, none of the participants had a history of polycystic ovary syndrome (PCOS) or an abnormal level of glucose or insulin.

### 3.2. The Infiltration and Accumulation of CD45^+^ Immune Cells Are Increased in Endometrium from Patients with EH

To investigate the general proportions of immunocytes in the endometrium of the three groups, flow cytometry analysis was used. As shown, the ratio of immunocytes in both the proliferative phase and EH was higher than that in the secretory phase (*p* < 0.05, *p* < 0.0001). In addition, the percentage of CD45^+^ immunocytes in EH patients was the highest among the three groups, up to 81.13% (Figure 1A,B) (Table 4). However, there was no significant difference in the proportion of neutrophils in the endometrium of the three groups (Figure 2A,B). These data suggest that the greater infiltration and accumulation of CD45^+^ immune cells occur in the endometrium in patients with EH.

### 3.3. There Are Increased Ratios of CD8^+^ Cells and Macrophages, and Decreased NK Cells in Endometrium from EH Patients

As shown, the total T cell proportions were similar in the three groups. The proportions of effective CD4^+^ T cells in the endometrium of proliferative phase and EH patients were higher than that in the secretory phase endometrium (*p* < 0.001, *p* < 0.05) (Figure 3 A–C); however, there was no significant difference in the proportion of CD4^+^ T cells between the proliferative phase and EH groups (Figure 3A–C). Interestingly, EH patients had the highest proportion of CD8^+^ cells, and proliferative phase patients had the lowest proportion of CD4^−^CD8^−^ T cells (*p* < 0.001, *p* < 0.01) (Figure 3A–C).

Subsequently, we found that the patients in the proliferative phase had higher numbers of CD3^+^CD56^−^NK cells than patients in the secretory phase (*p* < 0.01) and EH groups (*p* < 0.0001) (Figure 4A,B), and the patients in the secretory phase group had the lowest populations of CD3^+^CD56^+^NKT cells in the endometrium (*p* < 0.001, *p* <0.01) (Figure 4A,B).

### 3.4. Macrophages Are Largely Infiltrated and Increased in Endometrium from EH Patients

In the normal proliferative phase, immune cells are scarcely infiltrated and most of them are macrophages. It was reported that the number of macrophages was elevated significantly in the endometria of both EH and EC patients [4,14]. In order to identify the changes in macrophages, we performed FCM analysis. Of note, we observed that EH patients had the highest levels of CD14^+^ macrophages (nearly 80%), followed by patients in the secretory phase group (Figure 5A,B). These results suggest that macrophages represent the largest population of immune cells in the endometria of patients with EH.

### 3.5. HO-1 and Nrf2 Inhibitors Increase the Expression of Chemokines in Endometrial Epithelial Cells (EEC)

The Nrf2/HO-1 axis is closely associated with multiple gynecological cancers, such as ovarian cancer and endometrial cancer [15], due to its regulation of cell proliferation, metastasis, the immune response, etc. To explore the role of the Nrf2/HO-1 axis, we firstly analyzed the protein–protein interaction (PPI) network across chemokines, adhesion molecules and heme metabolism-related factors on the basis of the STRING database (Figure 6A). Among the related molecules in the PPI network, HMOX1/HO-1 and NFE2L2/Nrf2 might be important regulators for chemokines. As shown, stimulation with heme significantly up-regulated the mRNA levels of HO-1 and Nrf2 in EECs, suggesting that supplementation with heme can activate heme metabolism by increasing the expression of HO-1 and Nrf2 in EECs, especially at 25 μM (Figure 6B). To further analyze the potential role of heme metabolism in EECs, EEC cells were treated with a HO-1 inhibitor (Zinc Protoporphyrin) at 5 μM or a Nrf2 inhibitor (ML385) at 5 μM for 24 h, respectively, and then chemokines in EECs were detected by RT-PCR. We observed that the expression of CCL2, CCL3 and CXCL8 was increased in EECs treated with the HO-1 inhibitor (Zinc Protoporphyrin), and the expression of CCL3, CCL5 and CXCL8 was increased in those treated with the Nrf2 inhibitor (ML385) (Figure 6C,D).

### 3.6. Excess Heme Increases the Expression of Chemokines in EECs and Migration of Macrophages in Co-Culture System

Heme is involved in the differentiation and inflammatory activation of macrophages [16]. To investigate whether an increase in heme concentration would promote the expression of chemokines in EEC cells and the infiltration of macrophages, we treated EECs with 0, 12.5 and 25 μM of heme for 48 h, and then detected the expression of chemokines by PCR. As shown, the levels of CCL2, CCL3, CCL5 and CXCL8 were significantly increased after the treatment with heme at the concentration of 25 μM. Moreover, the expression of CCL2 and CXCL8 was raised when treated with 12.5 μM of heme (Figure 7A). In addition, EECs were pre-treated with 0, 12.5 and 25 μM of heme for 48 h and then co-cultured with macrophages for 24 h. In the chemotaxis assay, we found that EECs pre-treated with 12.5 and 25 μM of heme may increase the infiltration of macrophages (Figure 7B).

## 4. Discussion

Endometrial hyperplasia is a common gynecological disease [1], complicated by AUB. In our study, we found that the proportions of immune cell subsets in the endometria of EH patients with AUB were abnormal, especially regarding macrophages. Other diseases causing AUB, as outlined in the FIGO 2 classification system, are also related to macrophages. For example, a strong association between an increase in macrophages and adenomyosis has been demonstrated in the literature [17], despite there being no strong association between adenomyosis and EH/EC [18]. Moreover, it is reported that there is increased infiltration of macrophages in myoma, especially submucosal myoma [19]. This is possibly due to excess heme increasing the expression of chemokines and promoting the migration of macrophages.

The increasing population of immune cells may result in local tissue inflammation and is associated with the transformation of healthy tissue into cancerous tissue (such as EH, gastric cancer) [20]. In the normal endometrium, CD45^+^ leukocytes comprise up to 40% of the total cells in the pre-menstrual phase of the menstrual cycle [21]. It is noted that leukocytes increase in the secretory phase in preparation for the occurrence of menstruation, which is referred to as endometrial leukocyte infiltration [22,23]. Some evidence has suggested that some diseases complicated by AUB may be related to immunological disturbances, including EH [21]. Immune cell populations play complex roles in EH, which needs further study.

T cell subsets mainly include CD4^+^ T cells, CD8^+^ T cells, CD4^−^CD8^−^ T cells and NKT cells. CD3^+^ T cells can be detected throughout the menstrual cycle, but they only account for 1–2% of the total lymphomyeloid cells [24]. However, in our previous study, the proportion of CD3^+^ T cells in the endometria of EH patients was up to 12.5%, and the numbers of CD4^+^ T cells and CD8^+^ T cells also increased [4]. CD8^+^T cells are a key component of tumor-infiltrating cytotoxic lymphocytes (CTLs), regarded as a hallmark of the tumor immune response. NKT cells contain both markers of NK and T cells [25], identified as CD3^+^ CD56^+^. NKT cells produce IFN-γ to activate NK cells, T cells and macrophages, consequently of vital importance in regulating different immune responses and in protection from tumor growth and metastasis [26]. Consistent with previous studies, we found that the numbers of CD4^+^ T cells and NKT cells in proliferative phase patients and in EH patients were higher than those in the secretory phase, whereas the number of CD8^+^ T cells in EH was higher than that in the proliferative phase or secretory phase. It can be speculated that with the progression of EH, the abnormal immune environment led to the rise in the proportions of cytotoxic CD8^+^ T cells and helper CD4^+^ T cells, further resulting in the imbalance of T cells and accelerating the progression to EH.

Natural killer (NK) cells in the endometrium are called endometrial granular lymphocytes (EGL) [27], identified as CD56^+^ CD16^−^ [28]. It has been reported that NK cells are involved in irregular bleeding through destroying the integrity of blood vessels in the endometrium and altering its functions [29]. In this study, NK cells displayed different and varying trends from the proliferative phase to the secretory phase in AUB patients. In contrast, there was a reduction in EH. It was reported that NK cells differentiated toward a killer phenotype (CD56^+^ CD16^+^) by the stimulation of high levels of progestin in the study of Witkiewicz et al. [30], whereas NK cells were stimulated by high estrogen and low progestin in EH, possibly contributing to the reduction in NK cells.

Neutrophils are rarely tested in the normal endometrium. Only at the time prior to menstruation do the populations rapidly rise to 6–15% of the total cells [23]. Neutrophils play an important role in endometrial repair [31]. Although there is no marked difference in number during the proliferative phase or secretory phase in normal women [21], similarly to women with AUB or EH with AUB, their roles should not be ignored.

In the normal endometrium, T cells and NK cells represent the majority of leukocytes, followed by macrophages [32,33]. Macrophages can be detected throughout the menstrual cycle, with an increase in number starting from the proliferative phase [34], for the promotion of inflammatory endometrial destruction, repair and regeneration [23,35]. Resident immune cells, including macrophages, play a key role in immunity and homoeostasis in the endometrium [36]. The results of this study showed that the population of macrophages was the highest in EH. The activation of macrophages may be accompanied by the increased secretion of different inflammatory mediators and cytokines, such as tumor necrosis factor-a, interleukins (IL)-1 [37], platelet activating factors, vascular endothelial growth factor and angiogenesis factor, which may be associated with the development of EH, thus attracting our attention [38]. Therefore, in EH, the migration and differentiation of macrophages may also be changed by environmental factors and play a certain role in the occurrence and development of EH.

The main characteristic of EH is the irregular proliferation of endometrial epithelium cells, which is caused by the continuous stimulation of unopposed estrogen, which is mediated by inflammation and oxidative stress [39]. HO-1 is able to respond to electrophilic stimuli, including oxidative stress, which plays a key role in the pathogenesis of EH [40]. HO-1 may be involved in the growth, angiogenesis and metastasis of cancer cells, such as ovarian cancer and endometrial cancer [15]. HO-1 inhibitors, such as Zinc Protoporphyrin, have been confirmed to possess efficacy against cancer [41]. Nrf2 is the most essential activator of HO-1. Oxidative stress induces Nrf2 activation from resting conditions, and stimulates the Nrf2/antioxidant-responsive element (ARE)/HO-1 signaling pathway [42]. HO-1 is a rate-limiting enzyme in the breakdown of heme, the degradation of which produces biliverdin, carbon monoxide (CO) and iron (ferrous iron, Fe^2+^). Heme and heme metabolites participate in anti-apoptotic effects, promoting oxidation and inflammation, which lead to the development of EC [43]. However, some studies reported that HO-1 can ameliorate the EH induced by estrogen [44]. In this study, heme may have been a risk factor in EH patients with AUB, which may rely on macrophage infiltration. Existing studies have shown that heme may affect the functions of various immune cells, including macrophages [45,46]. Heme can induce pro-inflammatory cytokines or chemokines, possibly leading to neutrophil migration, macrophage infiltration and more. In addition, the CC chemokine receptors CCR2 and CCR5, and their cognate ligands (such as CCL2, CCL7 and CCL8 of CCR2, CCL5, CCL3 or CCL4 of CCR5), have been shown to regulate the recruitment of phagocytes in a variety of inflammatory diseases [47]. Heme can polarize macrophages either to the M1 subtype, via acting as TLR-4 and leading to TNF-α and IL-6 expression [16], or to the M2 subtype, via interacting with CD163 to induce an anti-inflammatory protective phenotype [48]; consequently, macrophages participate in the development of EH, as mentioned above. According to our data, heme-induced macrophage recruitment may be blocked by the use of inhibitors of HO-1 or Nrf2. These data suggest that targeting the heme/HO-1/Nrf2 axis may be a potential therapeutic strategy for EH and EC, and this type of therapy was shown to be successful in several animal models [45,49].

In conclusion, in the endometria of AUB patients, there are differences in the infiltration and residence of immune cells according to the physiological conditions, and there are more macrophages in the endometria of AUB patients with EH. AUB can lead to an increase in heme, causing oxidative stress damage by HO-1 and Nrf2, further promoting the transcription of chemokines such as CCL2, CCL3, CCL5, CXCL8, etc., and the recruitment of macrophages, and thus eventually exacerbating the progression of the disease (Figure 8). This provides potential early warning molecules and intervention targets for the treatment of EH. However, more intensive studies are still required to explore and demonstrate the involved mechanisms in greater detail. For example, the regulatory mechanism of heme metabolism and the detailed mechanism of heme involved in chemokine secretion still need to be studied.

## Figures and Tables

**Figure 1 biomolecules-12-00849-f001:**
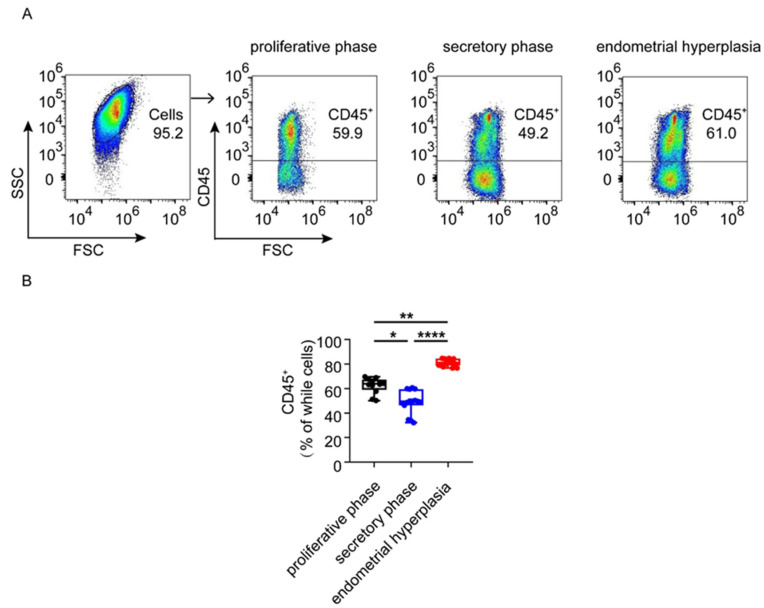
The numbers of local immunocytes were different in proliferative phase, secretory phase and endometrial hyperplasia endometrium of AUB patients. (**A**) Using flow cytometry gating strategy to distinguish CD45^+^ cells in proliferative phase, secretory phase and endometrial hyperplasia endometrium of AUB patients. (**B**) The proportion of CD45^+^ cells in whole cells. Data are presented as the median and the interquartile range. Statistical significance (Kruskal–Wallis test with Dunn’s multiple-comparison test): * *p* < 0.05, ** *p* < 0.01, **** *p* < 0.0001.

**Figure 2 biomolecules-12-00849-f002:**
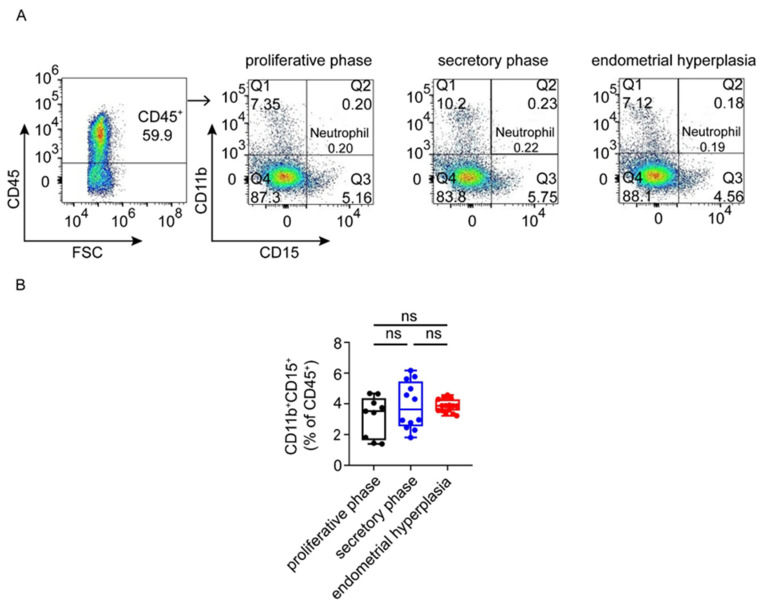
The proportion of neutrophils in proliferative phase, secretory phase and endometrial hyperplasia endometrium of AUB patients. (**A**,**B**) Gating strategy was used to distinguish neutrophils. CD45^+^ gate was first used to gate cells, followed by CD11b and CD15 gates. Graph shows the proportion of total CD45^+^CD11b^+^CD15^+^ neutrophils in proliferative phase, secretory phase and endometrial hyperplasia groups. Data are presented as the median and the interquartile range. Statistical significance (Kruskal–Wallis test with Dunn’s multiple-comparison test): ns, no significant difference.

**Figure 3 biomolecules-12-00849-f003:**
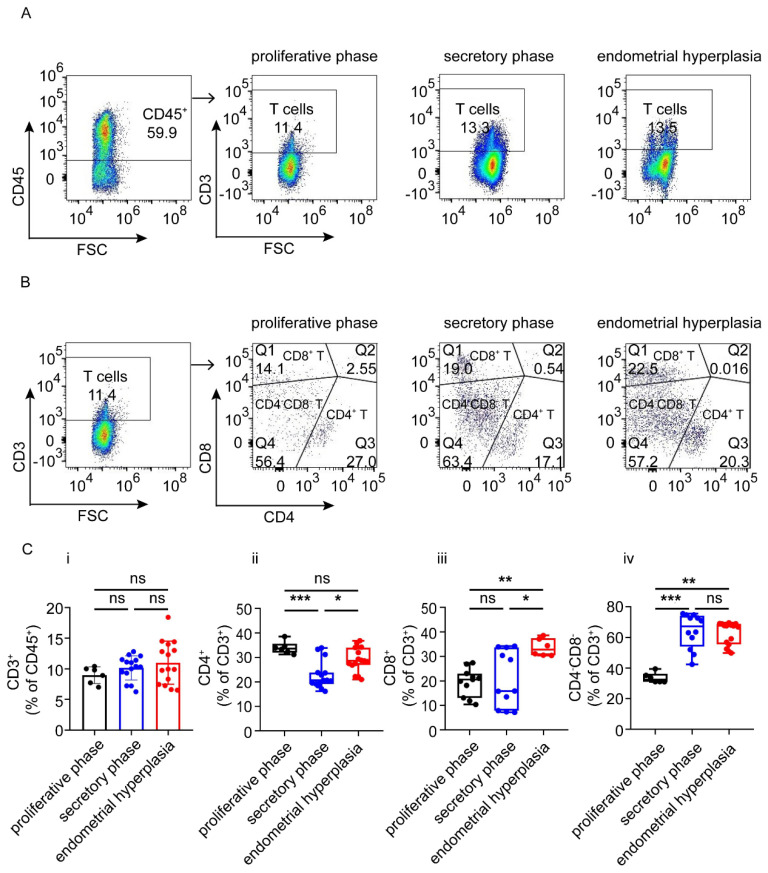
Different subtypes of T cells in proliferative phase, secretory phase and endometrial hyperplasia endometrium of AUB patients. (**A**) CD45^+^ gate, followed by CD3^+^ gate, was used to distinguish T cells. (**B**) Gating strategy was used to distinguish CD4^+^ T cells, CD8^+^ T cells and CD4^−^CD8^−^ T cells. (**C**) Graph shows the proportion of total T cells, CD4^+^ T cells, CD8^+^ T cells and CD4^−^CD8^−^ T cells in proliferative phase, secretory phase endometrium and endometrial hyperplasia endometrium of AUB patients. Data are presented as the mean ± standard error of the mean (**C.i**), or the median and the interquartile range (**C.ii**–**C.iv**). Statistical significance (one-way ANOVA with a Bonferroni multiple-comparisons test (**C.i**) or Kruskal–Wallis test with Dunn’s multiple-comparisons test (**C.ii**–**C.iv**)):, * *p* < 0.05, ** *p* < 0.01, *** *p* < 0.001, ns, no significant difference.

**Figure 4 biomolecules-12-00849-f004:**
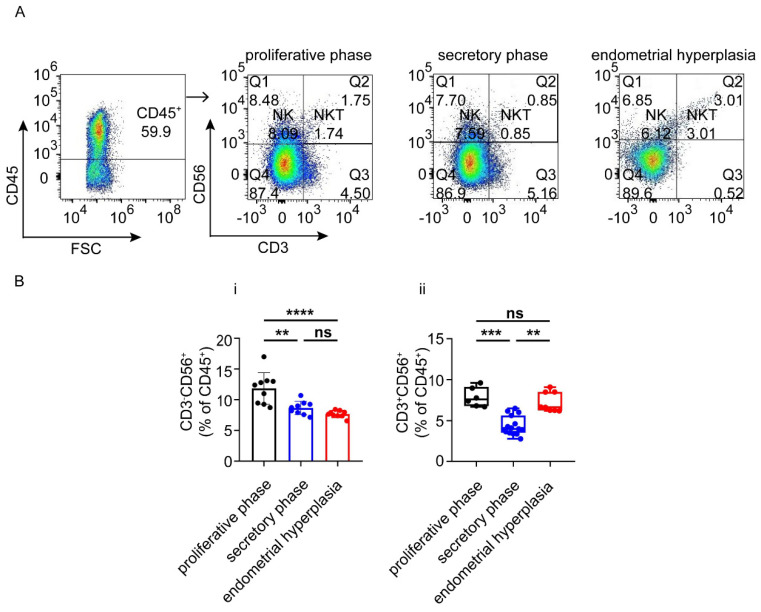
NK and NKT cells in proliferative phase, secretory phase and endometrial hyperplasia endometrium of AUB patients. (**A**) CD45^+^ gate, followed by CD3^+/−^CD56^+^ gate, was used to distinguish NK cells and NKT cells. (**B**) Graph shows the proportions of general NK cells and NKT cells in three groups. Data are presented as the mean ± standard error of the mean (**B.i**), or the median and the interquartile range (**B.ii**). Statistical significance (one-way ANOVA with a Bonferroni multiple-comparisons test (**B.i**) or Kruskal–Wallis test with Dunn’s multiple-comparisons test (**B.ii**)): ** *p* < 0.01, *** *p* < 0.001, **** *p* < 0.0001, ns, no significant difference.

**Figure 5 biomolecules-12-00849-f005:**
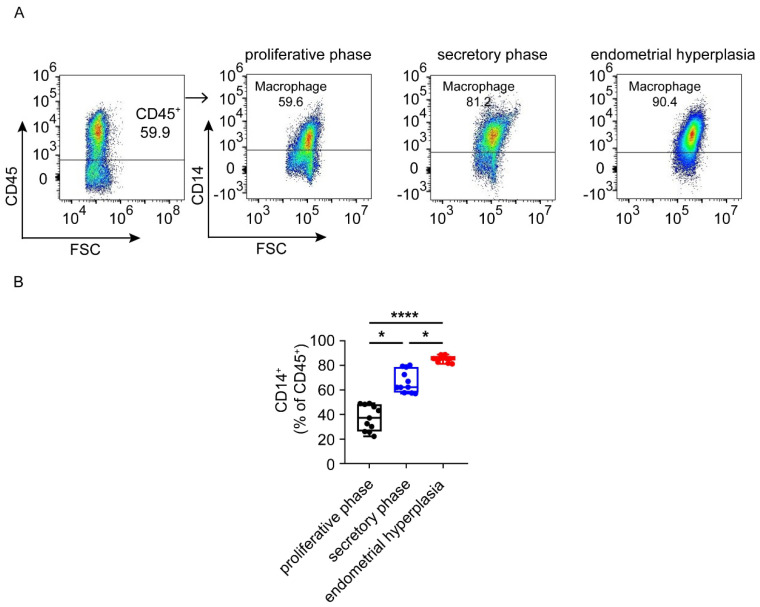
The proportion of macrophages in proliferative phase, secretory phase and endometrial hyperplasia endometrium of AUB patients. (**A**) CD45^+^ gate, followed by CD14^+^ gate, was used to distinguish monocytes. (**B**) Graph shows the proportions of general monocytes in three groups. Data are presented as the median and the interquartile range. Statistical significance (Kruskal–Wallis test with Dunn’s multiple-comparison test): * *p* < 0.05, **** *p* < 0.0001.

**Figure 6 biomolecules-12-00849-f006:**
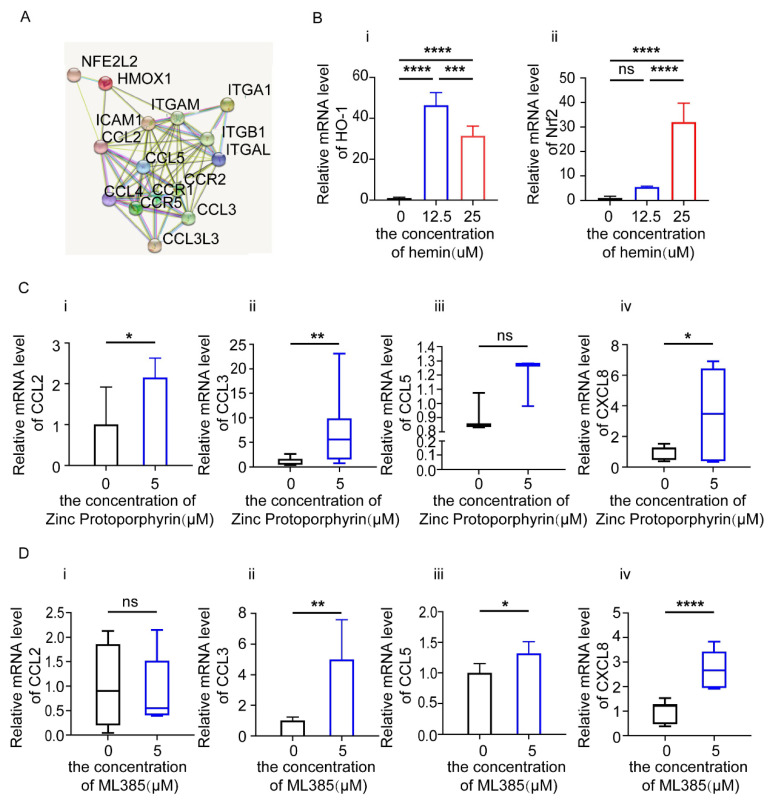
Blocking HO-1 and Nrf2 leads to an increase in chemokines by EECs. (**A**) The PPI network of chemokines, adhesion molecules and heme metabolism-related factors. (**B**) Relative mRNA expression of *HO-1* and *Nrf2* in EECs treated with Hemin at a concentration of 0 μM, 12.5 μM and 25 μM for 48 h. (**C**) Relative mRNA expression of *CCL2*, *CCL3*, *CCL5*, *CXCL8* in EECs treated with HO-1 inhibitor (Zinc Protoporphyrin) at the concentration of 5 μM for 24 h. (**D**) Relative mRNA expression of *CCL2*, *CCL3*, *CCL5*, *CXCL8* in EECs treated with Nrf2 inhibitor (ML385) at the concentration of 5 μM for 24 h. Data are presented as the mean ± standard error of the mean (**B.i**,**B.ii**,**C.i**,**D.ii**,**D.iii**), or the median and the interquartile range (**C.ii**–**C.iv**,**D.i**,**D.iv**). Statistical significance (one-way ANOVA with a Bonferroni multiple-comparisons test for three groups (**B.i**,**B.ii**), and t test (**C.i**,**D.ii**,**D.iii**) or Mann–Whitney U test (**C.ii**–**C.iv**,**D.i**,**D.iv**) for two groups): * *p* < 0.05, ** *p* < 0.01, *** *p* < 0.001, **** *p* < 0.0001, ns: no significance.

**Figure 7 biomolecules-12-00849-f007:**
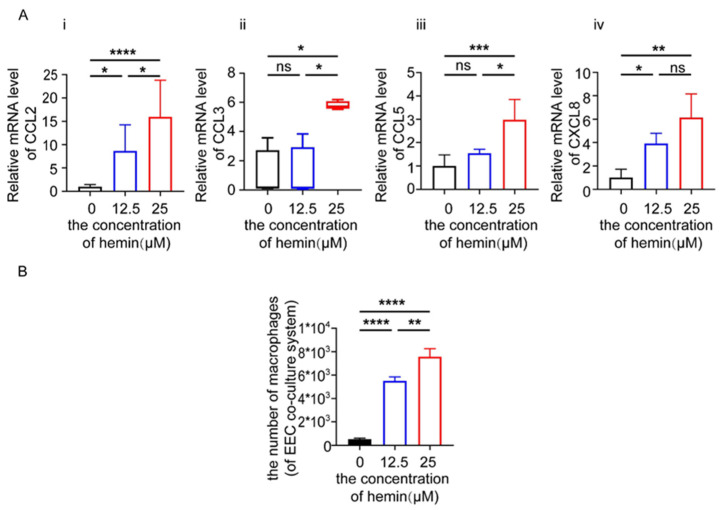
Heme increases the expression of chemokines in EECs and recruits macrophages. (**A**) Relative mRNA expression of *CCL2*, *CCL3*, *CCL5*, *CXCL8* in EECs treated with Hemin at the concentration of 12.5 μM and 25 μM for 48 h. (**B**) We illustrate the recruitment of macrophages co-cultured with EECs, which were untreated or treated with Hemin at the concentration of 12.5 μM or 25 μM for 48 h; we counted the number of recruited macrophages. Data are presented as the mean ± standard error of the mean (**A.i**,**A.iii**,**A.iv**,**B**), or the median and the interquartile range (**A.ii**). Statistical significance (one-way ANOVA with a Bonferroni multiple-comparisons test (**A.i**,**A.iii**,**A.iv**,**B**) or Kruskal–Wallis test with Dunn’s multiple-comparisons test (**A.ii**)): * *p* < 0.05, ** *p* < 0.01, *** *p* < 0.001, **** *p* < 0.0001, ns: no significance.

**Figure 8 biomolecules-12-00849-f008:**
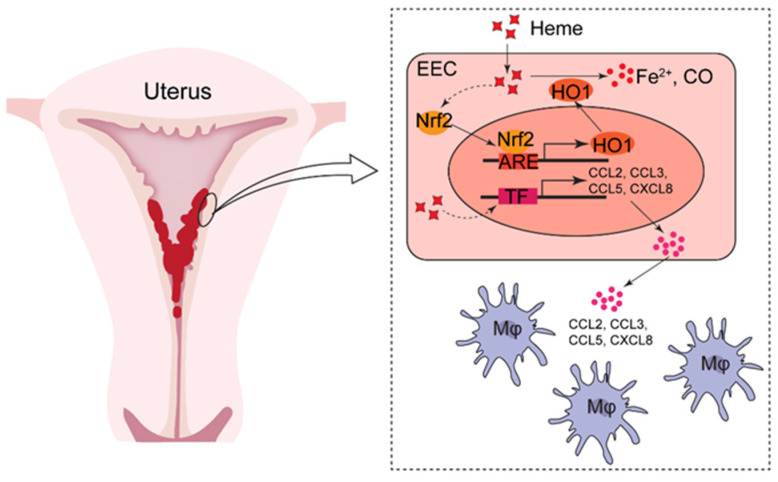
A schematic chart of excess heme in the migration and infiltration of macrophages from AUB patients with EH. In endometria of patients with EH complicated by abnormal uterine bleeding, there should be massive heme release. A certain level of heme activates the Nrf2/HO-1 axis, and further promotes heme metabolism. However, excess heme stimulates EECs to produce more chemokines (e.g., CCL2, CCL3, CCL5), possibly leading to the migration and infiltration of macrophages in the endometrium and accelerating the development of EH.

**Table 1 biomolecules-12-00849-t001:** Baseline Characteristics.

	Proliferative Phase	Secretory Phase	Endometrial Hyperplasia
**Cases**	15	15	15
**Pathology reports**	Proliferative period of the menstrual cycle	Secretory phase of the menstrual cycle	Endometrial hyperplasia
**Age (years)**	35.40 ± 6.98	39.0 ± 12.10	36.78 ± 2.83
**Endometrial thickness (mm)**	9.2 ± 4.96	8.2 ± 1.48	13.56 ± 1.06
**Clinical symptoms**	AUB	AUB	AUB
**Polycystic Ovary Syndrome (PCOS)**	Without	Without	Without
**Glucose level**	Normal	Normal	Normal
**Insulin level**	Normal	Normal	Normal

The data are shown as mean ± SEM; one-way ANOVA with a Bonferroni multiple-comparisons test was used to compare data available in groups and there was no significant difference among the three groups in age and endometrial thickness (mm); AUB: abnormal uterus bleeding.

**Table 2 biomolecules-12-00849-t002:** Antibodies for Flow Cytometry.

Antibody	Manufacturer	Cat No.
Allophycocyanin/Cyanine 7 (APC/Cy7) Anti-human CD45	BioLegend	368515
Phycoerythrin (PE)/ Cy7 Anti-human CD14	BioLegend	982510
Brilliant violent (BV) 605 Anti-human CD56	BioLegend	362537
Fluorescein isothiocyanate (FITC) Anti-human CD3	BioLegend	981002
PE Anti-human CD16	BioLegend	980102
PE Anti-human CD4	BioLegend	980804
PE Anti-human CD11b	BioLegend	982606
PE/ Cy7 Anti-human CD8	BioLegend	980910
PE/Cy7 Anti-human CD15	BioLegend	301923

Anti-CD45 antibodies, anti-CD3 antibodies, anti-CD4 antibodies and anti-CD8 antibodies were used to identify T cells, CD4^+^ T cells and CD8^+^ T cells. Anti-CD45 antibodies, anti-CD3 antibodies, anti-CD56 antibodies, anti-CD14 antibodies and anti-CD16 antibodies were used to identify macrophages, NK cells and NKT cells. Anti-CD45 antibodies, anti-CD11b antibodies and anti-CD15 antibodies were used to identify neutrophils.

**Table 3 biomolecules-12-00849-t003:** Gene primers for PCR.

Gene	Sequence (5’→3’)
*ACTB*	Forwards: GCCGACAGGATGCAGAAGGAGATCA
	Reverse: AAGCATTTGCGGTGGACGATGGA
*CCL2*	Forwards: TCGCTCAGCCAGATGCAATCAATG
	Reverse: AGATCACAGCTTCTTTGGGACACTTG
*CCL3*	Forwards: CATGGCTCTCTGCAACCAGTTCTC
	Reverse: CTGGCTGCTCGTCTCAAAGTAGTC
*CCL5*	Forwards: CTCGCTGTCATCCTCATTGCTACTG
	Reverse: TTGCCACTGGTGTAGAAATACTCCTTG
*CXCL8*	Forwards: CTCTCTTGGCAGCCTTCCTGATTTC
	Reverse: TTTGGGGTGGAAAGGTTTGGAGTATG
*HO-1*	Forwards: TGCCAGTGCCACCAAGTTCAAG
	Reverse: TGTTGAGCAGGAACGCAGTCTTG
*Nrf2*	Forwards: AGTCCAGAAGCCAAACTGACAGAAG
	Reverse: GGAGAGGATGCTGCTGAAGGAATC

**Table 4 biomolecules-12-00849-t004:** The proportion changes in immunocyte subtypes in different groups.

Immunocytes Subsets	Proliferative Phase	Secretory Phase	Endometrial Hyperplasia
**Immunocytes**	+++	+++ ↓ *	++++ ↑ **
**Macrophages**	+++	++++ ↑ *	++++ ↑ ****
**T cells total**	+	+ ^ns^	+ ^ns^
**CD4^+^**	++	+ ↓ ***	++ ↑ ^#^
**CD8^+^**	+	+ ↓ ^$^	++ ↑ **
**CD4^−^ CD8^−^**	+++	++++ ↑ ***	+++ ↑ **
**NK**	−/+	−/+ ↓ **	−/+ ↓ ****
**NKT**	−	− ↓ ***	− ↓ ^##^
**Neutrophils**	−	− ^ns^	− ^ns^

The summary of the proportion changes in main immunocyte subtypes in three groups of AUB patients. The data are expressed as the mean ± standard error of the mean or the median and the interquartile range. Statistical significance (one-way ANOVA with a Bonferroni multiple-comparisons test or Kruskal–Wallis test with Dunn’s multiple-comparison test): Percentage—mean percentage <5%, −/+: 5–10%, +: 10–20%, ++: 20–40%, +++: 40–60%, ++++: >60%. * *p* < 0.05, ** *p* < 0.01, *** *p* < 0.001, **** *p* < 0.0001 vs. the proliferative phase group; # *p* < 0.05, ## *p* < 0.01 vs. the secretory phase group; $ *p* < 0.05 vs. the endometrial hyperplasia group. ns, no significant difference.

## Data Availability

Not applicable.

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
