# Peer review of "Excess Heme Promotes the Migration and Infiltration of Macrophages in Endometrial Hyperplasia Complicated with Abnormal Uterine Bleeding"

_biomolecules, 2022, doi:10.3390/biom12060849_

Round 1
Reviewer 1 Report
In this manuscript by Ruan et al., the role of heme in macrophage migration in endometrial hyperplasia has been studied. I find this concept interesting and worth exploring to describe the pathophysiology of endometrial hyperplasia and associated bleeding in connection with inflammation.
I suggest correcting numerous grammar and spelling mistakes to improve the flow of the article, perhaps by using an external English editing service. Phrasing of some sentences is not suitable, e.g. “There must be a large amount of heme release” – there is no must. If there is a probable causative relationship between heme release and AUB, the authors can hypothesize or cite some evidence. I agree that there probably is heme release during bleeding, but this sentence sounds unscientific.
Multiple figure legends describe the statistical methods used for calculation of significance. However, these descriptions are too unclear. It seems one phrase keeps being copy-pasted into all figures, however I don’t see how one would use ANOVA or Kruskal-Wallis for these calculations. “The data are 231 expressed as the mean ± standard error of the mean or the median and the interquartile range. Statistical significance (one-way ANOVA with a Bonferroni multiple comparisons test or Kruskal-Wallis test with Dunn's multiple comparison test)”. Please provide the raw data for the graphs in Excel or Graphpad format so the reviewers can look into these calculations. Please provide explanation for – which statistical test was used for which figure panel.
Fig. 4 describes NKT cells, although the gating strategy depicts CD14 gating? Please correct this error.
Fig. 5 describes CD14+ CD45+ cells as macrophages which I disagree with. CD14 expression is also present in monocytes. How did the authors differentiate between monocytes and macrophages?
The inhibitors used against HO-1 and Nrf2 – do they have significant off-target effects?
Author Response
Dear editors:
Thank you very much for your letter dated June 9 enclosing the reviewers’ comments for our manuscript entitled “Excess heme promotes the migration and infiltration of macrophage in endometrial hyperplasia complicated with abnormal uterine bleeding”. We submit a revised manuscript in which the revisions have been labeled in red. The following are the responses to the reviewers’ concerning comments and suggestions about the manuscript.
We wish to take this opportunity to express our gratitude for your reconsideration of our paper for publication in your journal.
Reviewer: 1
Comments to the Author
- I suggest correcting numerous grammar and spelling mistakes to improve the flow of the article, perhaps by using an external English editing service. Phrasing of some sentences is not suitable, e.g. “There must be a large amount of heme release” – there is no must. If there is a probable causative relationship between heme release and AUB, the authors can hypothesize or cite some evidence. I agree that there probably is heme release during bleeding, but this sentence sounds unscientific.
Response: Thank you for your kind comments. We have carefully corrected numerous grammar and spelling mistakes, and got the manuscript edited by a professional editing service (https://www.mdpi.com/authors/english).
- Multiple figure legends describe the statistical methods used for calculation of significance. However, these descriptions are too unclear. It seems one phrase keeps being copy-pasted into all figures, however I don’t see how one would use ANOVA or Kruskal-Wallis for these calculations. “The data are 231 expressed as the mean ± standard error of the mean or the median and the interquartile range. Statistical significance (one-way ANOVA with a Bonferroni multiple comparisons test or Kruskal-Wallis test with Dunn's multiple comparison test)”. Please provide the raw data for the graphs in Excel or Graphpad format so the reviewers can look into these calculations. Please provide explanation for – which statistical test was used for which figure panel.
Response: Thank you for your helpful comments. First, the data were tested for normality and independence. For the two groups of data, according to whether they conform to the normal distribution, choose the t test (normal distribution) or Mann-Whitney U test (non-normal distribution). For multiple groups of data, one-way ANOVA test analysis was used for normal distribution, and Kruskal-Wallis test with Dunn's multiple comparison test was selected for non-normal distribution. Data were expressed as the mean ± standard error (SEM) of normally distributed data, and other forms were expressed as median and quartile. p<0.05 was used to indicate a statistically significant difference. We added detailed explanation in the legend. Additionally, we have provided the raw data for the graphs in Excel.
- Fig. 4 describes NKT cells, although the gating strategy depicts CD14 gating? Please correct this error.
Response: Thank you. We have corrected this error.
4.Fig. 5 describes CD14+ CD45+ cells as macrophages which I disagree with. CD14 expression is also present in monocytes. How did the authors differentiate between monocytes and macrophages?
Response: Thank you for your helpful comments. The main immunocytes were macrophages, mast cells, NK cells, T cells and B cells in the endometria according to the literature (doi: 10.4110/in.2015.15.1.16) , the proportion of monocytes is estimated to be very low. Besides, we carefully removed blood from the samples before endometrium tissue digestion.
5.The inhibitors used against HO-1 and Nrf2 – do they have significant off-target effects?
Response: Thank you for your helpful comments. We set the appropriate concentration according to the instructions and reports (doi:10.1021/acschembio.6b00651, doi:10.1186/1471-2407-8-197), and we also set concentration gradient for better study.

Reviewer 2 Report
thank you for giving me the opportunity to review this paper. The argument and methodology is good, however I would like to ask the authors for more details on the frequency of other diseases causing AUB as outlined in the FIGO 2 classification system For example, a strong association between increase of MP and adenomyosis has been demonstrated in literature (10.1093 / humupd / dmaa038), despite there is not a strong association between adenomyosis and EH/EC(10.1007 / s00404-020-05840-8)
Please include a paragraph listing strenghts and limitations of the study
Author Response
Dear editors:
Thank you very much for your letter dated June 9 enclosing the reviewers’ comments for our manuscript entitled “Excess heme promotes the migration and infiltration of macrophage in endometrial hyperplasia complicated with abnormal uterine bleeding”. We submit a revised manuscript in which the revisions have been labeled in red. The following are the responses to the reviewers’ concerning comments and suggestions about the manuscript.
We wish to take this opportunity to express our gratitude for your reconsideration of our paper for publication in your journal.
Reviewer: 2
thank you for giving me the opportunity to review this paper. The argument and methodology is good, however I would like to ask the authors for more details on the frequency of other diseases causing AUB as outlined in the FIGO 2 classification system For example, a strong association between increase of MP and adenomyosis has been demonstrated in literature (10.1093 / humupd / dmaa038), despite there is not a strong association between adenomyosis and EH/EC (10.1007 / s00404-020-05840-8)
Please include a paragraph listing strengths and limitations of the study.
Response: Thank you for your helpful comments. We have provided more details on the frequency of other diseases causing AUB and cited these references. In additionally, the strengths and limitations have been supplemented in the last paragraph in the section of Discussion.
